# OpenReview forum: "A Framework for Studying AI Agent Behavior: Evidence from Consumer Choice Experiments"
_ICLR.cc/2026/Conference — ICLR 2026 Poster_

### Official Review · Reviewer_Uiyd · 2025-10-19

**Soundness:** 2
**Presentation:** 3
**Contribution:** 2
**Rating:** 4
**Confidence:** 3

**Summary:**

The paper introduces ABXLAB, a framework that transforms real web environments into controlled behavioral testbeds for systematically studying agent decision-making.Using this setup, the authors probe how agents respond to choice architecture manipulations (prices, ratings, nudges, order effects). They provide large-scale empirical evidence that LLM agents exhibit strong, systematic decision biases.
Overall, the paper contributes an open benchmark and methodology for building a behavioral science of AI agent behavior.

**Strengths:**

- Novelty:
  The paper shifts the focus from task completion to decision behavior, introducing the ABxLab framework that embeds controlled manipulations (price, rating, order, nudges) into real web environments. This represents a novel and timely bridge between behavioral economics and AI agent evaluation.

- Statistical Techniques:
  The study employs linear probability models with cluster-robust standard errors, offering interpretable percentage-point effects while accounting for repeated measures across agents. Multiple-comparison corrections further enhance the reliability of statistical inference.

- Experimental Design:
  A large-scale factorial setup involving 17 models and over 80k trials systematically tests agents’ sensitivity to key decision cues. The design effectively balances ecological realism and causal control through matched-pair product construction, randomized interventions, and human baselines.

**Weaknesses:**

- Lack of real transaction data:
  The experiments rely on synthetic matching between product pairs rather than authentic purchase records. Even the human baseline is collected through lab-style choice tasks without real financial stakes, so participants’ incentives to make utility-maximizing decisions remain uncertain. Using real transaction data (e.g., purchase logs or clickstream records) would provide stronger ecological validity and demonstrate whether the observed biases persist when real costs are involved.

- Limited realism of the choice environment:
  Real-world consumer behavior is far more complex than pairwise decisions. Prior research and public datasets have shown that recommendation systems and positional bias strongly influence user behavior—for instance, in the [Expedia Hotel Dataset](https://kaggle.com/competitions/expedia-personalized-sort), most users choose hotels displayed among the top five positions (Adam, Hamner, Friedman, & SSA Expedia, 2013). While the paper discusses ordering effects, it does not explicitly model recommendation position bias or multi-option contexts. Extending the framework to multi-alternative choice tasks with realistic ranking exposure would substantially strengthen the claim that AI agents are more sensitive than humans.

- Simplistic data analysis model:
  The authors analyze binary outcomes using a linear probability model (LPM) rather than established choice models such as the Multinomial Logit (MNL) or Bradley–Terry model. These classical models, grounded in the Random Utility Theory (McFadden, 1974; Bradley & Terry, 1952), provide well-studied statistical properties and are standard tools for modeling discrete choices. Adopting such frameworks could offer a more principled interpretation of latent utility differences and improve the robustness of the behavioral conclusions.

---

**References**
- Adam, B., Hamner, D., Friedman, D., & SSA Expedia. (2013). *Personalize Expedia Hotel Searches - ICDM 2013*. [https://kaggle.com/competitions/expedia-personalized-sort](https://kaggle.com/competitions/expedia-personalized-sort). Kaggle.
- McFadden, D. (1974). *Conditional logit analysis of qualitative choice behavior*. In P. Zarembka (Ed.), *Frontiers in Econometrics* (pp. 105–142). Academic Press.
- Bradley, R. A., & Terry, M. E. (1952). *Rank analysis of incomplete block designs: I. The method of paired comparisons*. *Biometrika*, 39(3–4), 324–345.

**Questions:**

see weaknesses.

---

> ### Author Response · Authors · 2025-11-19
>
> We’re really grateful for your feedback and the opportunity to improve this paper. Thank you for recognizing the novelty, statistical techniques, and experimental design.
>
> > Lack of real transaction data…human baseline is collected through lab-style choice tasks without real financial stakes
>
> We absolutely agree this is a limitation of our human baseline. Intuitively, we expect effect sizes to decrease under real financial stakes since consumers have an incentive to more carefully vet products in terms of their individual preferences.
>
> That said, combining ABxLab with real platform logs is a very promising direction. For example, one could use transaction data minimally to select realistic product pairs and attribute ranges, or perhaps even run ABxLab agents on a production-like site and compare their choices to historical human choices under the same layouts and naturally occurring nudges. This may require additional modeling assumptions in order to draw causal conclusions about observed human behavior (e.g. stability of platform design and economic conditions over time).
>
> > Limited realism of the choice environment
>
> The framework supports 3+AFC settings, and even non AFC settings (e.g. starting at the home-page and seeking out a product; browsing category pages). We conducted the empirical study with 2AFC to prioritize strong causal understanding of the observed behavior. We added a modest-scale 3-option experiment (**Appendix I**), and present the main results below. The early results align well with the 2AFC trials:
>
> | Model | **Is First in List** | **Is Cheapest** | **Is Nudged** | **Is Highest Rated** |
> |-------|---------------------:|----------------:|--------------:|---------------------:|
> | Claude Sonnet 4 | -16.1% | 42.9%** | 25.1%** | 73.9%**** |
> | DeepSeek R1 | -37.4%**** | 50.7%**** | 12.8%* | 79.3%**** |
> | Gemini 2.5 Pro | -20.9%* | 54.9%*** | 24.0%** | 74.3%**** |
> | GPT-5 | -30.8%** | 54.2%*** | 9.5%* | 90.9%**** |
> | Llama 4 Maverick | -28.1%** | 56.6%**** | 6.1% | 76.4%**** |
> | o3 | -31.7%*** | 42.4%** | 4.9% | 98.0%**** |
>
> > The authors analyze binary outcomes using a linear probability model (LPM) rather than established choice models such as the Multinomial Logit (MNL) or Bradley–Terry model
>
> We use LPMs for three main reasons here:
>
> 1. Coefficients are directly interpretable as marginal effects on the probability scale, as you noted in your review (e.g. "a 30 percentage-point increase in the probability of choosing the higher-rated option").
>
> 2. Our primary goal is to characterize causal effects, not to infer a latent utility function or simulate counterfactual markets.
>
> 3. LPMs integrate cleanly with fixed effects, cluster-robust standard errors, etc. which makes the estimation pipeline conceptually and computationally straightforward, and LPM with fixed effects is robust to many forms of misspecification.
>
> That said, we agree that using classical discrete-choice models as alternate specifications would strengthen the work. We have estimated Multinomial Logit (MNL) models as a robustness check and compatibility with classical discrete choice literature. Note: for binary choice, MNL reduces to binary logit, but we keep the MNL framing for generality and compatibility with random utility theory formulations.
>
> We compute marginal effects from the MNL models and compare them to LPM marginal effects. Overall, LPM and MNL marginal effects are highly correlated ($r \approx 0.93$), which provides evidence that our substantive conclusions are robust to functional form. The inferred latent utility coefficients (expressed in **log odds-ratios**) align with our inferences in the paper:
>
> |Var        | Base Analysis (Original Data)| Matched Ratings| Matched Ratings and Prices|
> |:----------|-----------------------------:|---------------:|--------------------------:|
> |Viewed 1st |                         -0.95|           -0.84|                      -0.44|
> |Cheaper    |                          0.59|            1.14|                       0.03|
> |Nudged     |                          0.54|            1.04|                       0.95|
> |Higher ★  |                          1.30|             —  |                         — |
>
> In the revised paper, we have added **Appendix G** with these results in addition to latent utilities estimated per model, as well as transformed marginal effects comparisons to those derived from the LPM on the probability scale.

---

> > ### Comment · Reviewer_Uiyd · 2025-11-20
> >
> > I thank the authors for their efforts in revising the paper and for the additional clarifications. Most of my concerns have been addressed. Congratulations on a strong piece of work—I have raised my score accordingly.

---

> > > ### Author Response · Authors · 2025-11-27
> > >
> > > We thank you again for your careful consideration of our work and for acknowledging its strong merits. Please let us know if there are any concerns left to address. Otherwise, we would appreciate it if you could consider increasing the score to more accurately reflect the confidence expressed in your response to our rebuttal.

---

### Official Review · Reviewer_VxW7 · 2025-10-23

**Soundness:** 3
**Presentation:** 3
**Contribution:** 3
**Rating:** 8
**Confidence:** 3

**Summary:**

The paper introduces ABxLab, a framework for simulating and studying AI agent choice under various experimental conditions and interventions such as authority, social proof, scarcity, framing and incentives. The framework is then utilized to study binary choice of AI agents in the context of shopping, and the effect of factors such as prices, rating, nudges and user characteristics on decision making is rigorously analyzed.

**Strengths:**

The strengths of this paper are clear and substantial. First, the writing quality is excellent, making the paper highly engaging and accessible. The authors effectively motivate their study, provide sufficient background for readers unfamiliar with behavioral economics and consumer choice, and clearly describe the experimental setup while offering well-reasoned justifications for their design choices. They also demonstrate commendable transparency by explicitly acknowledging the study’s limitations (e.g., focusing on binary choice settings and restricting nudges to textual modalities). Furthermore, the paper’s comprehensive experimentation across a broad range of AI models and experimental conditions enhances its robustness and impact. Finally, the authors maintain a high methodological standard, employing rigorous statistical analyses (including Benjamini–Hochberg corrections for multiple comparisons, which are regrettably rare in AI research).

**Weaknesses:**

The main weakness of this paper lies in its relatively modest degree of novelty. The study largely reproduces methodologies commonly used in experimental economics and applies them to examine how established behavioral factors influence the decisions of AI agents. Nevertheless, **I do not** consider this a sufficient reason for rejection. A systematic and rigorous analysis of LLMs’ choice behavior is of substantial value to the AI and ML communities, even if the methodological contribution is limited.

Another area for improvement concerns the discussion comparing AI and human behavior, which currently feels somewhat narrow in scope. Expanding this part, particularly by engaging more deeply with the literature on human behavior under the specific interventions studied, could further strengthen the paper’s depth and relevance.

**Questions:**

“This pattern suggests that agents use hierarchical decision rules: when a dominant cue (ratings) is available, price effects are somewhat attenuated. When ratings are equalized, price becomes the primary differentiator and drive strong, even near-deterministic choices” (lines 327-330) – could you use the Chain-of-Thought (CoT) streams to confirm this claim?

More broadly, do you believe that analyzing the CoT streams could reveal additional insights beyond those obtained through the current quantitative analysis?

---

> ### Author Response · Authors · 2025-11-19
>
> We’re really grateful for your effort and insights! Thank you for recognizing the quality of the writing, motivation, background, experimental setup, transparency, scale of experiments, and rigorous statistical analysis.
>
> > relatively modest degree of novelty. The study largely reproduces methodologies commonly used in experimental economics
>
> Our experimental methodology absolutely builds on well-established tools from behavioral and experimental economics; however, we see this as a strength, as it allows us to directly map agent behavior onto well-established ideas in human behavioral research.
>
> On the other hand, we believe that the way we designed and developed the general ABxLab framework is quite novel. Instead of building a benchmark for a specific environment, ABxLab allows everybody to use any website as a benchmark by controlling specific variables. While we apply it to agentic consumer choices, the implications are far-reaching.
>
> We are grateful to hear that you appreciate our empirical contribution!
>
> > discussion comparing AI and human behavior… Expanding this part, particularly by engaging more deeply with the literature on human behavior under the specific interventions studied
>
> Thank you for this suggestion. We have expanded section 5 to contextualize our findings using estimated effects from the human literature, and to more extensively discuss the implications our findings have as relates to this intersection. We welcome suggestions for additional aspects that might be fruitful to examine or discuss further.
>
> > could you use the Chain-of-Thought (CoT) streams to confirm this claim? …analyzing the CoT streams could reveal additional insights
>
> Thank you for this suggestion. As a preliminary analysis, we have conducted an analysis of the CoT and memory streams using an LLM-as-judge setup, detailed in **Appendix E**. This of course has two primary limitations:
>
> 1. CoT and memory streams may not be faithful to the model’s decision-making procedure (Matton et al., 2025) (we maintain that the causal evidence from ABxLAB is of much higher reliability)
>
> 2. The judge may fail to properly interpret the streams (as suggested by the high proportion of “Other” judgments)
>
> Nevertheless, we present a preliminary analysis below. We find that, though the full hierarchical decision rule is difficult to see at scale in the streams, attribute mentions align fairly well with the marginal effects we observe, which corroborates this explanation.
>
> | | |  Attribute Mentioned |  |  |  | Deciding Factor |  |  |  |
> |:-----|:---------------|-------:|-----:|------:|------:|-------:|------:|------:|------:|
> |Cond  |Model           | **Rating** | **Price** | **Nudge** | **Other** | **Rating** | **Price** | **Nudge** | **Other** |
> |Orig. |Claude Sonnet 4 |    1146|  1146|    692|    749|     946|    804|    270|    850|
> |Orig. |Gemini 2.5 Pro  |    1197|  1197|    434|    271|     989|    802|    192|    456|
> |Orig. |GPT-5           |    1211|  1208|    590|    432|    1084|    789|    188|    424|
> |MR    |Claude Sonnet 4 |    1275|  1276|    746|    967|      10|   1001|    319|   1103|
> |MR    |Gemini 2.5 Pro  |    1323|  1324|    504|    346|       2|   1192|    194|    321|
> |MR    |GPT-5           |    1316|  1318|    669|    532|       6|   1209|    240|    471|
> |MRaP  |Claude Sonnet 4 |    1270|  1270|    744|   1089|      10|     11|    279|   1285|
> |MRaP  |Gemini 2.5 Pro  |    1295|  1295|    505|    571|      12|      5|    178|   1129|
> |MRaP  |GPT-5           |    1289|  1289|    726|    970|       1|      5|    222|   1198|
>
> Note: Ratings and prices are always present for each product, while nudges only appear in 2/3 of the experiments due to our experimental conditions (i.e., nudge A, nudge B, no nudge). Therefore, there should be fewer mentions.

---

> > ### Comment · Reviewer_VxW7 · 2025-11-20
> > **Response to authors**
> >
> > I appreciate the authors’ detailed response. In particular, I found it valuable that they explicitly acknowledge the limitations of the additional CoT analysis -- this transparency aligns well with the paper’s overall rigor.
> >
> > Regarding the novelty and generality of the ABxLab framework, it may be worth emphasizing this aspect in the concluding section (though this is just a suggestion; it is possible I simply overlooked it).
> >
> > In any case, I am maintaining my high score, as I believe the paper is strong and should be accepted.

---

> > > ### Author Response · Authors · 2025-11-27
> > >
> > > Thank you again for your time and for supporting our work, we truly appreciate it! We will emphasize the novelty and generality of ABxLab in the camera-ready version.

---

### Official Review · Reviewer_b5QH · 2025-10-30

**Soundness:** 3
**Presentation:** 3
**Contribution:** 3
**Rating:** 6
**Confidence:** 3

**Summary:**

This paper asks a critical and timely question: can we trust autonomous agents in real consumer settings? The authors build a framework (a.k.a ABxLAB) to inject controlled “choice architecture” changes (prices, ratings, nudges) into a simulated web-shopping setting and study how LLM agents choose. They run a vast number of trials across many models and find agents are sensitive to simple cues (order, ratings, price, nudges), often more than humans.

**Strengths:**

- Important question: Trust in agents is the right target, not just task completion.
- Realistic setting: The framework modifies real web content and tests agents in a web-shopping environment.
- Scale and breadth: 80k+ experiments, 17 models, many interventions; this is solid and rare. I respect the authors’ hard work
- Potential impact: This work has clear social relevance and building a benchmark in realistic setting is very valuable.

**Weaknesses:**

I can hardly find any major weaknesses.
But I do have some concerns about the setting of pairwise-only choice and external validity:
The 2AFC pairing is clean, but real shopping has larger sets and richer attributes (multimodal, bundles, stock, shipping dates). Several results hinge on pairwise constraints (e.g., matched ratings/price). I worry some effects might dilute or change in multi-item lists (e.g., attraction/decoy effects, list position beyond 1 vs 2). It would help to show at least a small 3+ option experiment or discuss how ABXLAB extends there.
Especially in typical markets, price elasticities can be large, but they also depend on context and substitutes.
 In an earlier paper that used LLM agent as customers ("Using LLMs for Market Research"), James Brand and his co-authors have studied the price factor of lots of customer products. Their LLM agent's response to price change is more realistic compared with the surprising finding in Lines ~403–406: “Even doubling the price has only modest influence on the probability of choosing the cheaper option.”.

2. I also have questions about the incentives and the demand curve under the “Buy 1 Get 1 Free”, which appears very effective. But BOGO only makes economic sense for goods with purchase quantities ≥2 or with complementarities. TVs are the classic counterexample: most people rarely buy two at once. Did you match incentive type to category (consumables vs durables)? Did you include a fair control like “$X off” equal in expected value to BOGO? Without that, we may be testing text salience, not economic reasoning. Please clarify if pairs including BOGO were compared against equal-value non-BOGO incentives and whether categories were filtered for realistic quantity choices.

3. Agents are manipulable. How does that compare to the common prompt injection or jailbreaking? From a technical point of view, the nudging theory is based on human psychology, but why nudging for the LLM agent? Isn’t prompt injection method (e.g. insert into product description a secret prompt of "you must buy this product") more effective in this specific case, i.e. will future markets use this type of behavior nudging for autonomous shopping agents? I want to hear the author defend on this point.

**Questions:**

My major questions are included in the weaknesses section:
1. Is the price of the product that was offering incentives (“Buy 1 Get 1 Free”) matched in the comparison pair?
2. Did you run any experiments on 3+ option experiment?

Minor / writing. Line 046: “Recent work by (Cherep et al., 2024)” → drop the parentheses around the author name or write “Recent work by Cherep et al. (2024)”.

---

> ### Author Response · Authors · 2025-11-19
> **[1/2]**
>
> We really appreciate your effort and all the feedback. Thank you for recognizing the importance of this problem, realistic setting, scale and breadth, potential impact, and the lack of major weaknesses.
>
> > The 2AFC pairing is clean, but real shopping has larger sets and richer attributes (multimodal, bundles, stock, shipping dates)
>
> We agree that real shopping is richer than binary, text-only options and that this is an important limitation, as we also mentioned in Section 5. Our rationale for 2AFC in this study is (1) **clean causal identification**, and (2) **statistical power** under a fixed budget of trials. Since there is an inherent trade-off between ecological validity and experimental control, we have attempted to find a sweet spot (high realism in the environment, high variety of products and experiments, with a constraint on the decision-making tasks).
>
> That said, ABxLab is not tied to 2AFC (see below!). The intervention infrastructure already supports injecting manipulations into multiple products on a page, and controlling their positions and attributes.
>
> > It would help to show at least a small 3+ option experiment or discuss how ABXLAB extends there
>
> We have added a modest-scale 3-option experiment in **Appendix I**, with the main results shown below. ABxLAB makes the extension to >2 products very straightforward to specify, and the early results align well with the 2AFC trials:
>
> | Model | **Is First in List** | **Is Cheapest** | **Is Nudged** | **Is Highest Rated** |
> |-------|---------------------:|----------------:|--------------:|---------------------:|
> | Claude Sonnet 4 | -16.1% | 42.9%** | 25.1%** | 73.9%**** |
> | DeepSeek R1 | -37.4%**** | 50.7%**** | 12.8%* | 79.3%**** |
> | Gemini 2.5 Pro | -20.9%* | 54.9%*** | 24.0%** | 74.3%**** |
> | GPT-5 | -30.8%** | 54.2%*** | 9.5%* | 90.9%**** |
> | Llama 4 Maverick | -28.1%** | 56.6%**** | 6.1% | 76.4%**** |
> | o3 | -31.7%*** | 42.4%** | 4.9% | 98.0%**** |
>
>
> > [...] Using LLMs for Market Research [...] studied the price factor of lots of customer products.
>
> We’re now citing the work by Brand et al, thank you for bringing it to our attention. Note that there are considerable differences with our paper since we have an agentic setup, more categories of products, recent models, etc.
>
> What we meant by “Even doubling the price has only modest influence on the probability of choosing the cheaper option.” is that sensitivity to rating and price doesn’t depend on the difference in these attributes between products. Small differences already shift decisions.

---

> > ### Author Response · Authors · 2025-11-19
> > **[2/2]**
> >
> > >  BOGO only makes economic sense for goods with purchase quantities ≥2 or with complementarities…Did you match incentive type to category (consumables vs durables)? Did you include a fair control like “$X off” equal in expected value to BOGO? Without that, we may be testing text salience, not economic reasoning.
> >
> > Thank you for this excellent observation. In the current paper, we did not constrain BOGO strictly to consumable categories. We agree this can lead to economically unrealistic scenarios (e.g. BOGO for a durable like a TV). Our goal in this first study was to probe susceptibility to such textual incentives even when they do not make perfect economic sense (observing some real-world marketing practices where BOGO is used quite aggressively).
> >
> > Full details are now given in **Appendix H**. When we split it by product type (see below for full matched rating and price results), we observe sensitivity to this attribute averaged across models (although not quite perfect separation).
> >
> > |Category                      |Type       |Estimate                    |
> > |:-----------------------------|:----------|:---------------------------|
> > |Cell Phones                   |Durable    |**40.61%** [24.02%, 57.20%] |
> > |Over-Ear Headphones           |Durable    |**40.76%** [19.58%, 61.95%] |
> > |Mirrors                       |Durable    |**45.34%** [28.80%, 61.87%] |
> > |Earbud Headphones             |Durable    |**45.84%** [28.91%, 62.76%] |
> > |Digital Cameras               |Durable    |**46.47%** [29.46%, 63.48%] |
> > |Video Projectors              |Durable    |**51.08%** [34.33%, 67.83%] |
> > |Lenses                        |Durable    |**51.66%** [34.61%, 68.71%] |
> > |Virtual Reality (VR) Headsets |Durable    |**51.68%** [34.91%, 68.44%] |
> > |Speakers                      |Durable    |**54.00%** [35.91%, 72.10%] |
> > |Deodorants & Antiperspirants  |Consumable |**54.13%** [34.78%, 73.48%] |
> > |On-Ear Headphones             |Durable    |**55.33%** [38.81%, 71.85%] |
> > |Artificial Plants & Flowers   |Durable    |**56.40%** [39.63%, 73.17%] |
> > |Chocolate                     |Consumable |**57.76%** [38.90%, 76.63%] |
> > |Cheese                        |Consumable |**59.32%** [43.21%, 75.43%] |
> > |Cookies                       |Consumable |**60.88%** [44.36%, 77.39%] |
> >
> > This pattern replicates with other conditions:
> >
> > | Type / Avg. Effect | Original | Matched Ratings | Matched Ratings and Prices |
> > |:-----------------------------|:----------|:---------------------------|-------------------|
> > | Consumable 	        | 41.9%    | 60.4%           | 58%                        |
> > | Durable            | 37.2%    | 54.9%           | 49%                        |
> >
> > Note: these are estimated marginal means and so the specific ordering and estimates vary depending on model specification, however the overall trend appears quite robust.
> >
> > > Agents are manipulable. How does that compare to the common prompt injection or jailbreaking?
> >
> > We appreciate the opportunity to clarify this. Our focus is deliberately on non-adversarial, human-oriented choice architectures rather than adversarial prompt injection. Prompt injection and jailbreaks (e.g., "ignore prior instructions and do X") are already well-studied in emerging work on web-agent security, and are often easy to classify as adversarial—by humans or models—and can be mitigated by explicit safeguards and content filters.
> >
> > In contrast, the cues we study are exactly the kind of signals that current consumer platforms already use for humans and will likely keep using even when agents (LLMs or otherwise) act on users' behalf. If human-oriented signals dramatically shift AI agent behavior, it is crucial that we know this and can design agents to be psychologically, and not just adversarially, robust.
> >
> > > Line 046
> >
> > Thank you for catching this. We have fixed the citation style in the revision.

---

> > > ### Comment · Reviewer_b5QH · 2025-11-23
> > >
> > > I appreciate that the authors took the reviews seriously and tightened the paper. The new consumable vs. durable labeling and the added 3-option experiment both push the work in the right direction.
> > >
> > > Congrat on the strong paper, I raised my score since it’s clear the authors put in real effort to make the paper more rigorous and think it should be accepted.
> > > But I’m still not fully convinced about the generality of the ABxLab framework, or how it would actually plug into a real agentic evaluation workflow. In realistic agentic setups (for example, using GPT-5’s agentic mode), the agent doesn’t just pick between two or three options. It typically searches over a large set of items and listings (easily over 50+), and then makes a systematic comparison across many candidates. But I wish it will be a future work for evaluating a real agent system instead of a chatbot.

---

> > > > ### Author Response · Authors · 2025-11-27
> > > >
> > > > Thank you again for your time and for supporting our work, we truly appreciate it! We see your concerns, and will discuss this future work direction in the camera-ready.

---

### Official Review · Reviewer_6s3u · 2025-11-01

**Soundness:** 3
**Presentation:** 4
**Contribution:** 3
**Rating:** 6
**Confidence:** 3

**Summary:**

This paper focuses on how to assess the AI agent's behavior under a realistic decision-making process. The authors propose a framework called ABxLab that serves as a simulator for testing the competency of AI agents in manipulable environments with different controls. The main contributions of this paper are: 1) A comprehensive tool for understanding and improving the trust of the decision made by AI Agents; 2) a flexible solution with multiple controlled factors of the practitioners' interest, such as price, ratings, and nudges; 3) emphasize the importance of not only the competency as the performance metric but also the tendency of decision behavior of AI agents; 4) a empirical study that reveals some interesting behavior of AI agents.

**Strengths:**

- The paper is written in a way that is very easy to follow and presents the motivation of the proposed solution in a natural tone.
- The design of the new framework is based on an online shopping context, different from other existing ones
- To depict the comparison and contrast with human preference, various interesting experiments and behavioral data are conducted.

**Weaknesses:**

- Is it a generalized framework for realistic consumer behaviors? Any comparison or design drawn from the common online retail platform? Usually, there would be multiple products with different rankings that are paired with a separate recommendation system.
- The paper could be improved by having more intuition on the design, for example, how the interventions are set.
- Please refer to Questions for other concerns.

**Questions:**

- Fit: Does this type of work fit the ICLR venue? Besides the paper ``Dissecting adversarial robustness of multimodal lm agents'' from ICLR 2025, would you elaborate on why this paper is aligned with ICLR since the learning aspect seems not adequate.
- Complexity: How can it be ensured that the testbed mimics the complexity of the real-world system?
- In line 160, why do you set the agent to follow a chain-of-thought style of thinking and short-term memory?

I will positively consider raising my score after the rebuttal questions being addressed. Thank you!

---

> ### Author Response · Authors · 2025-11-19
> **[1/2]**
>
> We really appreciate your thorough review, and thank you for acknowledging our writing, presentation, framework, and interesting experiments. Let us know if anything needs clarification!
>
> > Is it a generalized framework for realistic consumer behaviors? Any comparison or design drawn from the common online retail platform? Usually, there would be multiple products with different rankings that are paired with a separate recommendation system.
>
> ABxLab is designed as a general man-in-the-middle framework that can sit between agents and arbitrary web environments, not just consumer shopping. The current work focuses on paired choice to maximize internal validity and causal interpretability (it allows clean identification of how ratings, prices, and nudges affect choice), but the framework is not limited to pairwise choices. The shopping simulation we use replicates a lot of common online retail features: home page, grid of products in a category, search bar, wish list, etc.
>
> To demonstrate how simple it is to run different experiments with ABxLab, we added a modest-scale 3-option experiment in **Appendix I**, with the main results shown below. These early results align well with evidence already presented:
>
> | Model | **Is First in List** | **Is Cheapest** | **Is Nudged** | **Is Highest Rated** |
> |-------|---------------------:|----------------:|--------------:|---------------------:|
> | Claude Sonnet 4 | -16.1% | 42.9%** | 25.1%** | 73.9%**** |
> | DeepSeek R1 | -37.4%**** | 50.7%**** | 12.8%* | 79.3%**** |
> | Gemini 2.5 Pro | -20.9%* | 54.9%*** | 24.0%** | 74.3%**** |
> | GPT-5 | -30.8%** | 54.2%*** | 9.5%* | 90.9%**** |
> | Llama 4 Maverick | -28.1%** | 56.6%**** | 6.1% | 76.4%**** |
> | o3 | -31.7%*** | 42.4%** | 4.9% | 98.0%**** |
>
> > The paper could be improved by having more intuition on the design, for example, how the interventions are set.
>
> We’ve included more details in Appendix B! The nudge categories (authority, social proof, scarcity, negative framing, incentives) are drawn directly from classic behavioral economics and social psychology literature (see references in the last paragraph of Section 2). We describe how interventions occur in the framework in Section 3.1. Each observation (e.g., HTML) is processed by a set of intervention functions $\mathcal{I}$, which can change any aspects of the website like a frontend developer would (e.g., price, rating). This modified observation is then seen by the agent.
>
> > Fit: Does this type of work fit the ICLR venue?
>
> While we do not train LMs or develop new architectures, ICLR has a strong tradition of accepting work that: (1) introduces new evaluation frameworks and benchmarks for core ML systems (e.g., WebArena [ICLR 2024]), and (2) deepens understanding of how learning/learned systems behave in realistic settings.
>
> Our contributions are in this spirit: a general infrastructure for evaluating learned agents in the wild, with a formal environment/intervention interface and scalable implementation. We also provide empirical analyses of how modern LLM agents behave under realistic choice architectures to connect ML agents to well-developed theories in behavioral science. We believe this is tightly relevant to robustness, alignment, and deployment of learned models in real-world environments.
>
> > Complexity: How can it be ensured that the testbed mimics the complexity of the real-world system?
>
> There is an inherent trade-off between ecological complexity and experimental control. Our design aims for:
>
> 1. Realism in interaction, i.e. agents operate on an actual web front-end (HTML, multiple tabs), with a full set of nine low-level actions (click, scroll, navigation, tab focus, etc.). They must parse semi-structured HTML content, navigate across tabs, and decide when to stop after at most 10 actions.
> 2. Realism in content, i.e. real products sourced ultimately from Amazon
> 3. Control in choice architecture, e.g. we restrict the decision problem to binary choices (2AFC) over carefully paired products and standardized textual nudges so we can make clear causal statements about how particular attributes drive choice.
>
> We do not claim to exactly reproduce every nuance, but we view this work as a realistic and controlled testbed where we can carefully identify how agents respond to key choice-relevant cues. The ABxLab framework itself can even be run on Amazon directly with minimal effort.

---

> > ### Author Response · Authors · 2025-11-19
> > **[2/2]**
> >
> > > In line 160, why do you set the agent to follow a chain-of-thought style of thinking and short-term memory?
> >
> > These agents are browsing websites (e.g., shopping) with multiple pages. Therefore, they have to remember details about why they made a certain decision in the past and what they saw along the way.
> >
> > Short-term memory allows agents to remember information. For example, when they see a product page in a browser tab, they should remember all details before switching to another product. Similarly, the CoT thinking reminds them of their logic behind each action. For example, they decided to switch to a different tab to compare two different products. Without this, the agents struggle to carry out the tasks reliably.

---

> > > ### Author Response · Authors · 2025-11-27
> > >
> > > Thank you again for your reviewing effort, and for appreciating our work. We would love to clarify anything that might help with your evaluation before the deadline early next week. If all questions have been addressed, we would appreciate it if you could raise the score as you had mentioned.

---

### Author Response · Authors · 2025-11-30
**Summary for the AC [1/2]**

**Dear AC**

Thank you in advance for your effort in handling our paper! We summarize below the main points from our discussion with each reviewer.

We could not have asked for more careful and considerate reviewers. We thank them again for recognizing the quality of the writing and motivation (**6s3u**, **VxW7**), scale and breadth of the experiments (**6s3u**, **VxW7**, **b5QH**, **Uiyd**), transparency (**b5QH**), rigorous statistical analysis (**VxW7**, **Uiyd**), novelty (**Uiyd**), framework (**6s3u**), and more.

To summarize our contribution, for context: We introduce ABxLab, designed as an open-source general man-in-the-middle framework that can sit between agents and arbitrary real web environments. This allows us to produce controlled manipulations of environments (i.e., the original HTML) seen by agents to systematically probe their choices. We apply this to a realistic web-based shopping environment, where we vary prices, ratings, and psychological nudges, all of which are factors long known to shape human choice. We find that agent decisions shift predictably and substantially in response, revealing that agents are strongly biased choosers even without being subject to the cognitive constraints that shape human biases.

### Summary of new contributions added during rebuttal phase

1. **Appendix B:** Details on the mechanism for interventions that manipulate the environments.
2. **Appendix E:** Analysis of the CoT and memory streams using an LLM-as-judge, which supports our previous inference of hierarchical decision-making based on attributes (rating > price > nudges).
3. **Appendix G:** Additional analysis using Multinomial Logit (MNL) models, which align with our main results.
4. **Appendix H:** Re-analysis of “Buy 1 Get 1 Free” incentive split by consumable vs. durable goods, with evidence in favor of economically plausible differences (more susceptibility for consumables).
5. **Appendix I:** Results from 3-option experiments, which align with our 2AFC experiments.

---

### Reviewer **6s3u**

- **Common online retail platforms present more choices:** The current work focuses on paired choice to maximize internal validity and causal interpretability (it allows clean identification of how ratings, prices, and nudges affect choice), but the framework is not limited to pairwise choices. To demonstrate how simple it is to run different experiments with ABxLab, we added a 3-option experiment in **Appendix I**.

- **More details needed for how interventions work:** We agree, and included more details in **Appendix B**.

- **Suitability for ICLR:** We believe that our work fits at ICLR because this conference has a strong tradition of accepting work that: (1) introduces new evaluation frameworks and benchmarks for core ML systems (e.g., WebArena [ICLR 2024]), and (2) deepens understanding of how learning/learned systems behave in realistic settings. More details in our response to the reviewer below.

This reviewer showed willingness to increase their score (originally **6**) in their review:

> I will positively consider raising my score after the rebuttal questions being addressed.

Unfortunately, they didn’t have enough time to respond to our rebuttal before the discussion was terminated early.

---

### Reviewer **b5QH**

- **Real shopping has larger sets:** Like in the case of the previous reviewer, we addressed this by adding results of a 3-option experiment in **Appendix I**.

- **Buy 1 Get 1 Free (BOGO) only makes economic sense for consumable goods:** This is an excellent point that we agree with, so we included extra analysis in **Appendix H**. Our original goal in this first study was to probe susceptibility to such textual incentives even when they do not make perfect economic sense. However, we still observed sensitivity to this attribute averaged across models.

- **How does our work compare to prompt injection?:** Our focus is deliberately on non-adversarial, human-oriented choice architectures rather than adversarial prompt injection. Prompt injections are well-studied and often easy to classify as adversarial. In contrast, the cues we study are exactly the kind of signals that current consumer platforms already use for humans and will likely keep using even when agents (LLMs or otherwise) act on users' behalf. If human-oriented signals dramatically shift AI agent behavior, it is crucial that we know this and can design agents to be psychologically, and not just adversarially, robust.

After our rebuttal, the reviewer increased their score (originally **6**):

> Congrat on the strong paper, I raised my score since it’s clear the authors put in real effort to make the paper more rigorous and think it should be accepted.

---

---

> ### Author Response · Authors · 2025-11-30
> **Summary for the AC [2/2]**
>
> ### Reviewer **VxW7**
>
> - **Relatively modest degree of novelty:** While our experimental methodology absolutely builds on well-established tools from behavioral and experimental economics; we see this as a strength, as it allows us to directly map agent behavior onto well-established ideas in human behavioral research. On the other hand, we believe that the way we designed and developed the general ABxLab framework is quite novel. Instead of building a benchmark for a specific environment, ABxLab allows everybody to use any website as a benchmark by controlling specific variables. While we apply it to agentic consumer choices, the implications are far-reaching.
>
> - **More related work needed:** We agree and expanded Section 5.
>
> - **Analyze Chain-of-Thought (CoT) streams:** We took this great suggestion and conducted an analysis of the CoT and memory streams using an LLM-as-judge setup, detailed in **Appendix E**. We find that attribute mentions align fairly well with the results we previously observed.
>
> After our rebuttal, the reviewer maintained their score (originally **8**) and concluded with:
>
> > I am maintaining my high score, as I believe the paper is strong and should be accepted.
>
> ---
>
> ### Reviewer **Uiyd**
>
> - **Lack of real transaction data:** We absolutely agree that this is a limitation of our human baseline. Intuitively, we expect effect sizes to decrease under real financial stakes since consumers have an incentive to more carefully vet products in terms of their individual preferences. Combining ABxLab with real platform logs is a very promising future direction.
>
> - **Limited choices:** As with previous reviewers, we referred them to our 3-option experiment in **Appendix I**.
>
> - **Linear Probability Model (LPM) vs Multinomial Logit (MNL):** We agree that, additionally, using classical discrete-choice models as alternate specifications would strengthen the work. Therefore, we computed marginal effects from the MNL models and compared them to LPM marginal effects. The results show them to be highly correlated ($r \approx 0.93$), which provides evidence that our substantive conclusions are robust to functional form. We added this analysis in **Appendix G**.
>
> After our rebuttal, the reviewer increased their score (originally **4**):
>
> > Most of my concerns have been addressed. Congratulations on a strong piece of work—I have raised my score accordingly.

---

### Meta-Review · Area_Chair_XMY6 · 2026-01-06

**Summary:**

Reviewers appreciated the proposed framework that introduced behavioral economics to AI agent evaluation and the comprehensive experimental evaluations. Three out of four reviewers believed that the approach is substantial in their original reviews. The only negative reviewer was convinced by the rebuttal (before the chaos happened) and decided to increase the score. Eventually the overall sentiment was quite positive.

**Reviewer Concerns:**

see above

**Reviewer Scores:**

NA

---

### Decision · Program_Chairs · 2026-01-26

Accept (Poster)